# Electroencephalogram Features of Perioperative Neurocognitive Disorders in Elderly Patients: A Narrative Review of the Clinical Literature

**DOI:** 10.3390/brainsci12081073

**Published:** 2022-08-13

**Authors:** Xuemiao Tang, Xinxin Zhang, Hailong Dong, Guangchao Zhao

**Affiliations:** Department of Anesthesiology and Perioperative Medicine, Xijing Hospital, Fourth Military Medical University, Xi’an 710032, China

**Keywords:** postoperative neurocognitive disorders, electroencephalogram, functional connectivity, general anesthesia

## Abstract

Postoperative neurocognitive disorder (PND) is a common postoperative complication, particularly in older patients. Electroencephalogram (EEG) monitoring, a non-invasive technique with a high spatial–temporal resolution, can accurately characterize the dynamic changes in brain function during the perioperative period. Current clinical studies have confirmed that the power density of alpha oscillation during general anesthesia decreased with age, which was considered to be associated with increased susceptibility to PND in the elderly. However, evidence on whether general anesthesia under EEG guidance results in a lower morbidity of PND is still contradictory. This is one of the reasons that common indicators of the depth of anesthesia were limitedly derived from EEG signals in the frontal lobe. The variation of multi-channel EEG features during the perioperative period has the potential to highlight the occult structural and functional abnormalities of the subcortical–cortical neurocircuit. Therefore, we present a review of the application of multi-channel EEG monitoring to predict the incidence of PND in older patients. The data confirmed that the abnormal variation in EEG power and functional connectivity between distant brain regions was closely related to the incidence and long-term poor outcomes of PND in older adults.

## 1. Introduction

The development of medical services with the ageing population has become an important issue worldwide. Each year, over 19 million patients aged ≥65 years undergo surgery under general anesthesia [1]. Perioperative neurocognitive disorders (PND), a type of common neurological complication, occurred in up to 50% of elderly patients undergoing major or high-risk operations, such as hip-fracture repair and cardiac surgery [2,3]. The category of PND includes: (1) cognitive decline diagnosed before operation (described as neurocognitive disorders); (2) postoperative delirium (POD); (3) delayed neurocognitive recovery to 30 days; and (4) postoperative neurocognitive disorder up to or beyond 12 months [4]. PND not only prolongs hospital stays and increases hospital costs, but also significantly increases mortality and the incidence of dementia [5]. Hence, neuroscientists and anesthesiologists have long been committed to developing a series of reliable, fast, and safe techniques to provide an early warning of neurological damage within the perioperative period both accurately and sensitively, so as to prevent or ameliorate the risk of PND.

The electroencephalogram (EEG) is a kind of non-invasive technique, which accurately characterizes the dynamic changes in brain function with a high spatial–temporal resolution. In the past few decades, processed EEG (pEEG) monitors based on the electrical activity of the frontal cortex have been widely used in clinical practice [6]. pEEG indices are quantitative parameters generated by translating the EEG signal into an index of anesthetic depth ranging between 0 (isoelectric EEG) and 100 (fully awake) [7], such as the bispectral index (BIS) and patient state index (PSI). However, evidence from recent clinical studies has shown that the intraoperative monitoring of the depth of anesthesia via a commercial dimensionless index cannot effectively identify the pathological change in the brain [4]. EEG characterizes the multi-dimensional information of the brain state, including power spectrum, complexity, and functional connection between different brain regions, which can reflect the function of the whole brain. We suggest that more comprehensive information generated by the raw data of multi-channel EEG will contribute to finding the biomarkers of PND that have been ignored in previous studies. For example, as we need to recognize the impairment of brain function, we could characterize the disintegration of EEG signals from different regions to account for the breakdown of neural networks [8].

In this review, we discuss the recent evidence regarding the use and analysis of intraoperative and postoperative EEG patterns and summarize how EEG changes in elderly patients predict the occurrence or risk of PND (Table 1 and Table 2), with the aim to provide more insights into the pathophysiology of perioperative neurocognitive decline.

## 2. Intraoperative EEG Features and PND

### 2.1. Processed Electroencephalography Indices

A large amount of work has investigated whether the pEEG-guided dosing of general anesthetics can improve postoperative outcomes. For instance, in the Cognitive Dysfunction after Anesthesia (CODA) trial, Chan et al. [9] aimed to determine whether the use of BIS could decrease the incidence of POCD. A total of 921 patients (≥60 years old) undergoing major non-cardiac surgery were randomly assigned to receive either BIS-guided anesthesia, where anesthesiologists were instructed to target a BIS value between 40 and 60 during general anesthesia, or routine care, where BIS was monitored but not revealed to anesthesiologists. Postoperative cognitive decline at 3 months was found to be reduced in the BIS-guided group compared with the routine care group. A meta-analysis of 13 trials identified that pEEG-guided anesthesia could reduce the incidence of POD [40]. Similarly, Punjasawadwong et al. [41] found that EEG-guided anesthesia, such as BIS, ameliorated POD, POCD and their consequences in patients undergoing non-cardiac surgery with moderate-quality evidence. Patients included in this meta-analysis underwent a variety of procedures, from ambulatory surgery to major non-cardiac surgery. Previous studies about the relationship between the depth of anesthesia and PND have focused on comparing BIS-guided general anesthesia with routine care; however, the sub-study of the Anaesthetic depth and complications after major surgery (BALANCED) trial explored whether lighter general anesthesia could decrease the incidence of POD. They included 547 patients 60 years and older, undergoing elective surgery expected to last at least 2 h, who were randomized to receive BIS target 50 (light general anesthesia) or BIS target 35 (deep general anesthesia). POD was detected in 19% of the patients in the BIS target 50 group, compared with 35% in the BIS target 35 group (*p* = 0.010). The main finding was that light general anesthesia could decrease the incidence of POD in the elderly [10].

Many randomized controlled trials (RCTs) and meta-analyses reported that pEEG-guided care was able to decrease the incidence of POD or POCD, mainly depending on the reduction in anesthetic exposure and duration of EEG suppression [9,42]. However, recently, the Electroencephalography Guidance of Anesthesia to Alleviate Geriatric Syndromes (ENGAGES) trial [11] did not support the use of pEEG monitoring to reduce the risk of POD. In the ENGAGES trial, 1232 patients aged 60 years or older scheduled for major surgery under inhaled anesthesia were randomized to receive either EEG-guided anesthetic administration or standard anesthetic care. The incidence of POD was not significantly different between the two groups (26% vs. 23%; difference = 3%; 95% CI, −2.0% to 8.0%; *p* = 0.22). This result showed that the pEEG guidance of anesthesia in elderly adults did not reduce the incidence of POD. The ENGAGES trial has admitted having several problems, including the lack of a clinically relevant difference in general anesthetic concentration between groups (0.69 vs. 0.80 MAC). At the same time, the median time difference between groups with EEG suppression was small (7 vs. 13 min). In addition, the prevalence of POD was higher than in other trials because the patients were highly susceptible to developing POD. Therefore, pEEG monitors probably do not prevent POD in highly susceptible patients, but this does not mean that EEG-based anesthesia guidance does not prevent POD. Recently, Sun et al. [43] published an updated meta-analysis of five trials with a sum of 3612 patients; the result did not support the preventive effect of pEEG-guided anesthesia on POD. However, they performed a thorough sensitivity analysis which showed that the ENGAGES trial accounted for 30% of the data in the meta-analysis and can be viewed as an outlier in relation to the other trials. Although protocols using pEEG indices to titrate drug effect have had discordant results on PND rates, intraoperative pEEG indices have been investigated to avoid POD and other sorts of PND in elderly patients.

pEEG indices could be affected by anesthetics, age, and preoperative cognitive status in older patients, so it was difficult to consider these indices as independently predictive for neuro-outcomes such as POD [6]. In a large single-center cohort with 4699 patients aged >30 years receiving general anesthesia with volatile anesthetics, Ni et al. [44] reported that the elderly received higher age-adjusted minimum alveolar concentration (MAC) fractions of volatile anesthetics but displayed increased BIS values compared with the young (*ρ* = 0.15; 95% CI: 0.12–0.17; *p* < 0.001). The BIS values in aged patients were paradoxically higher than those in younger patients. Moreover, BIS values increased with the administration of nitrous oxide or ketamine [45,46], underestimating the true depth of anesthesia. Therefore, BIS monitoring might be an inaccurate measurement of the effect of anesthetics in elderly adults which may lead to an incorrect exaggeration during general anesthesia, especially in the elderly.

### 2.2. EEG Spectral Analysis

Due to the limitation of pEEG in anesthesia monitoring, researchers have attempted to utilize unprocessed EEG to monitor and assess brain activity. Recent guidelines from Perioperative Quality Initiative recommend that anesthesiologists learn to interpret basic EEG, such as the raw waveform and spectrogram, to decrease the incidence of PND [47]. In this section, we review the recent findings of important EEG features related to the occurrence of PND, including decreased alpha power, burst suppression, and EEG connectivity and complexity.

#### 2.2.1. Alpha Band Activity

The frequencies of raw EEG signal always range from 0.3 to 45.0 Hz and can be divided into five bands according to amplitude and frequency [48]. The alpha band (8–13 Hz) is of particular interest because it changes with age during general anesthesia. Purdon et al. [49] investigated the influence of age on EEG spectral distribution during general anesthesia. The EEG spectra in young (18–38 years old) and elderly (70–90 years old) patients that received propofol or sevoflurane anesthesia were compared throughout a 2 min period of stable anesthetic maintenance. The results showed that EEG power across all frequency bands declined significantly with increasing age during the stable maintenance of anesthesia, and the change in alpha power was much more significant than that in other frequencies. The age-related decrement in alpha power might be derived from the decline in central synaptic density, changes in dendritic dynamics, and the reduction in neurotransmitters within the cerebral cortex [50,51]. These findings also reflect the underlying alterations in the properties of neural circuits with regard to alpha oscillation generation. Alpha power is thought to originate from thalamocortical electrical transmission [52], and participates in the modulation of arousal, attention, and other important cognitive functions. As a result, alpha power has been the subject of several studies investigating the biomarkers of neurocognitive decline during the perioperative period.

#### 2.2.2. Alpha Power and Preoperative Cognitive Decline

Preoperative cognitive function is associated with distinct EEG changes in the alpha band during anesthesia induction. Cartailler et al. [12] initially established an association between transient amplitude decreased (TAD) and cognitive status in a prospective study involving 38 patients (>50 years old) undergoing orthopedic surgery or neuroradiology intervention. The authors assessed cognitive function before surgery using the Montreal Cognitive Assessment method (MoCA). The target concentration of propofol (TCI) was set to 5 μg/mL during the induction of general anesthesia. They found that lower preoperative MoCA scores were linked to a rapid TAD increase and an alpha power decrease measured during the first 10 min of propofol-induced general anesthesia. Furthermore, the slope of TAD was related to the patient’s cognitive performance. Measuring the TAD slope at the onset of general anesthesia could indicate whether patients have cognitive impairment in the preoperative setting.

Elderly patients with cognitive decline in the preoperative setting have also been demonstrated to exhibit less intraoperative frontal alpha power during the maintenance of anesthesia. In an observational prospective cohort study, Giattino et al. [13] enrolled 50 patients aged 60 and older undergoing non-cardiac, non-neurologic surgery under propofol or isoflurane anesthesia They assessed preoperative neurocognitive function using their well-established neurocognitive test battery, and then measured the average alpha power using 32-channel EEG or BIS monitoring. The analysis of the EEG recordings from 15 patients who underwent 32-channel EEG and from 35 patients underwent single-channel prefrontal EEG using the BIS monitoring identified a significant association between intraoperative frontal alpha power and preoperative cognitive index score (*rs* = 0.593, *p* = 0.022; *rs* = 0.338, *p* = 0.047, respectively). In all cases, intraoperative frontal alpha power was correlated with preoperative cognitive performance in 50 elderly patients anesthetized with propofol or isoflurane. However, such a relationship between frontal alpha power and preoperative cognitive function was not observed in other EEG frequency bands. Gutiérrez et al. [14] demonstrated that patients with lower MoCA scores exhibited lower alpha power and slower alpha peak frequency, which is consistent with the possibility that elderly patients with poor cognitive function receive an overdose of brain anesthesia compared to cognitively more robust patients. Similarly, Touchard et al. [15] included 42 patients who had a preoperative cognitive assessment using MoCA with EEG information collected under a propofol-based general anesthesia, and they found that patients with cognitive decline had a lower alpha power and target TCI. Both articles demonstrate that preoperative cognitive status is associated with the sensitivity of the effect of general anesthetic. Thus, patients with some degree of cognitive impairment are more sensitive to the effect of general anesthetic, expressed as a greater slowing of EEG activity at the same anesthetic dose than in a cognitively more robust patient. Unable to show significant frontal alpha power under general anesthesia might be used as an intraoperative electrophysiological phenotype of preoperative cognitive decline. Meanwhile, from a prospective observational cohort study of 38 patients aged 65 and over undergoing elective surgery, as reported by Koch et al. [16], patients with reduced cognitive function presented a lower intraoperative alpha power. In contrast, the baseline of frontal alpha power in the preoperative period showed no correlation with preoperative overall cognitive function. Although cognitive decline was associated with reduced frontal alpha power in the resting brain [17], the authors found no correlation between baseline alpha power and preoperative cognitive performance. These studies indicate that age-dependent changes in cerebral activity are more sensitive for predicting cognitive function under general anesthesia than some other status. In summary, anesthesia-induced frontal alpha activity was related to poorer preoperative cognitive performance in elderly patients.

#### 2.2.3. Alpha Power and Postoperative Cognitive Decline

Intraoperative alpha power was also correlated with postoperative neurocognitive function. In order to investigate the potential intraoperative EEG patterns correlated with POD and subsyndromal delirium (PSSD), Gutierrez et al. [53] conducted an observational exploratory study, which included 36 patients aged 60 and older scheduled for elective major abdominal surgery. POD or PSSD was measured by the CAM instrument after surgery, and EEG signals were collected by a 16-channel EEG before and during the administration of inhaled anesthetics. They found that subjects in the POD/PSSD group compared with the control group had reduced intraoperative absolute (4.4 ± 3.8 dB vs. 9.6 ± 3.2 dB, *p* = 0.0004) and relative alpha power (0.09 ± 0.06 vs. 0.21 ± 0.08, *p* < 0.0001). Additionally, relative alpha power was able to identify patients with reduce cognitive function with an area under the curve of 0.90 (CI 0.78–1, *p* < 0.001). This phenomenon suggested that the thalamocortical feedback mechanism associated with intraoperative alpha power was disrupted in patients with POD or PSSD [54]. The susceptibility of POD or PSSD was unmasked under the effects of general anesthetics. The reduced intraoperative alpha power could be utilized as a novel EEG biomarker to identify patients with a high risk of POD or PSSD.

In summary, such works have validated that decreased alpha power during the maintenance and induction of anesthesia may predispose patients to cognitive decline during the perioperative period, forming the basis for the suggestion that the “manipulation of alpha power during general anesthesia may prove to enhance early neurocognitive recovery in older adults” [55,56]. It is also suggested that EEG-guided dosing of general anesthetics to generate the desired alpha oscillation would be able to minimize the incidence of PND. Further studies should identify the clinical utility of the unprocessed EEG and its spectrogram to individualize the dose of anesthetic required for elderly patients.

### 2.3. Burst Suppression (BS)

Under the deep state of unconsciousness induced by anesthetic drugs, burst suppression is composed of alternations between isoelectricity and brief bursts of EEG signal [57]. In actuality, evidence on the occurrence of burst suppression and PND is controversial.

#### 2.3.1. Burst Suppression Is Detrimental for POD

Some studies have demonstrated that intraoperative EEG burst suppression and duration predicted POD. In a prospective clinical trial, Fritz et al. [18] included 727 patients receiving volatile-based general anesthesia. They assessed POD using CAM-ICU on postoperative days 1 through 5 and measured the burst suppression obtained from BIS monitoring. POD was more likely to develop in patients who experienced more suppression(χ2(4) = 25; *p* < 0.0001). A correlation was demonstrated between intraoperative burst suppression and POD in a multivariate regression analysis after adjusting for potential confounders (OR = 1.22; 99%CI, 1.06–1.40; *p* = 0.0002). Similar findings were reported by Momeni et al. [19] in a prospective study that enrolled 1515 patients scheduled for cardiac surgery or transcatheter aortic valve implantation (TAVI). Recently, a prospective observational study of 81 subjects undergoing cardiac surgery [20] showed that delirious patients stayed in a burst suppression state for much longer than non-delirious patients (107 min, IQR [47;170] vs. 44 min, IQR [11;120], *p =* 0.018). Pedemonte et al. [21] included 159 subjects aged >60 years scheduled for cardiac surgery with CPB. They evaluated POD using the long version of the CAM and measured burst suppression by analyzing EEG data in the spectral and time-series domains. POD was identified to be more prevalent in subjects with CPB burst suppression compared with the subjects without CPB burst suppression (25% vs. 6%). In a multivariate logistic model, burst suppression was found to be an indicator of POD (OR = 4.1,95%CI 1.5–13.7, *p* = 0.012). A retrospective observational study by Lele et al. [22], which included 112 subjects aged between 20 and 88 years old undergoing spine instrumentation surgery with total intravenous anesthesia (TIVA), similarly adds to our understanding. They found that the incidence of intraoperative burst suppression was higher in subjects who developed POD (100% vs. 66.7%; *p* = 0.03, RR: 1.5, 95% CI: 1.3–1.7). Above all, intraoperative burst suppression activity could indicate a deep stage of anesthesia, which may trigger the development of POD in the elderly.

Furthermore, POD is more prevalent in patients who experienced burst suppression at a relatively low volatile anesthetics dose, which was identified by Ackland et al. [58] in a pivotal discussion about the ENGAGES trial. A retrospective analysis of 618 adult subjects who underwent general anesthesia for elective surgery reported by Fritz et al. [23] also found that EEG suppression at a lower inhaled anesthetics dose resulted in a higher incidence of POD in an adjusted logistic regression model (adjusted OR: 2.13,95% CI 1.24–3.65, *p* = 0.006). Besides, this phenotype is likely to be a more sensitive indicator of POD than the duration of intraoperative burst suppression [59].

In summary, the above studies all demonstrated that EEG suppression is an independent risk factor for POD, and that anesthesia-induced burst suppression may be a potential mechanism resulting in postoperative cognitive impairment. However, it is unclear whether there is a causative correlation between intraoperative burst suppression and POD, or burst suppression simply reflects the patient’s susceptibility to POD. If burst suppression is the cause of POD, then EEG-guided low-dose anesthesia that decreases the incidence of intraoperative burst suppression could reduce the risk of POD. However, if burst suppression solely reflects a patient’s potential susceptibility to POD, then we should identify patients who are susceptible to intraoperative burst suppression early in order to prevent or ameliorate this condition.

#### 2.3.2. Burst Suppression Is Protective or Has No Effect on POD

Contrary to the adverse impact of burst suppression, some studies have suggested that monitoring burst suppression has no effect on postoperative cognitive function, or that burst suppression may be protective. In the ENGAGES study, in contrast to findings from the preceding studies, the authors reported they did not find a statistically significant difference in the incidence of POD between the two groups [11], despite the duration of both EEG suppression (time with BIS < 40 decreased from 60 min to 32 min) and anesthetic dose (median end-tidal anesthesia concentration reduced from 0.8 to 0.7 MAC) being reduced. Similarly, the results of the Anesthetic Depth and Postoperative Delirium Trial-2 (ADAPT-2) trial did not find that decrease in the duration of EEG suppression provided any benefits with regard to the attenuation of the risk of POD (17% vs. 20%, *p* = 0.53) [24]. Previous studies have linked intraoperative burst suppression to POD, yet modestly shortening the time with EEG suppression had not been demonstrated to decrease the incidence of POD.

In a retrospective observational cohort of 105 subjects aged over 68 years undergoing major non-cardiac surgery, Deiner et al. [25] reported that the duration of burst suppression was shorter in subjects who developed POCD (35 [88] minutes vs. 96 [131] minutes; *p* = 0.04). The results of this study are similar to a study conducted by Valchanov et al. [60], who found that deep state of anesthesia was protective against POCD. Recently, Shortal et al. [61] studied 27 healthy humans anesthetized with isoflurane 1.3 MAC for 3 h. They recorded the duration of burst suppression and then assessed cognitive impairment using independent cognitive tests. Contrary to current assumptions, in healthy volunteers, the amount of burst suppression had no correlation with the degree of impairment in cognitive performance at emergence. That is, in healthy adults, EEG suppression was not a significant determinant of cognitive impairment after general anesthesia, but whether this finding holds true in elderly patients requires further verification.

Different algorithm values have been utilized to replace the examination of visual burst suppression [62]. Given the differences in the methods used to quantify burst suppression in the observational studies and RCTs described above, it is difficult to draw a consistent conclusion when comparing the results of these studies. Further research is needed to explore the causality between intraoperative EEG burst suppression and PND.

#### 2.3.3. Burst Suppression and Alpha Band Activity

The anesthesia-induced frontal alpha oscillation was not independent of burst suppression. Shao et al. [26] studied burst suppression and a decline in anesthesia-induced frontal alpha power from a trial where 155 patients were under propofol and sevoflurane anesthesia. By using logistic regression, they observed an important association between a reduction in the intraoperative frontal alpha power and an increased propensity of burst suppression. Moreover, they discovered that for each decibel reduction in alpha power, the likelihood of experiencing burst suppression increased by 1.33-fold. The correlation between burst suppression and alpha power was similarly reported by Plummer et al. [27]. The authors analyzed a sum of 138 EEG recordings obtained during general anesthesia from their database and found that reduced EEG power between 7.8 and 22.95 Hz for the maintenance phase of anesthesia was linked to the incidence and duration of burst suppression. As reduced alpha and beta power preceded the onset of burst suppression, their results imply that intraoperative EEG oscillations within the alpha and beta range may be further developed as EEG biomarkers for burst suppression. An intraoperative lower alpha power is significantly correlated with a higher probability for burst suppression and, therefore, an increased risk of PND. More deeply, the EEG alpha power reflects the integrity of cortical and cognitive circuits, and burst suppression is likely to occur in patients with damaged cortical and cognitive circuits. Hence, both intraoperative burst suppression and lower alpha power can be used to characterize brain vulnerability.

### 2.4. EEG Connectivity

The concept of functional connectivity refers to the temporal coincidence of spatially distant neurophysiological events and is generally assessed through functional magnetic resonance imaging (fMRI), EEG or magnetic electroencephalography (MEG) [63]. Recordings of EEG provide a more direct measure of brain dynamics. Directed connectivity could provide additional information about the direction of information flow. Disturbances in the organization of functional brain networks are seen in anesthetic-induced unconsciousness. Ku et al. [64] enrolled 18 patients receiving propofol or sevoflurane anesthesia and demonstrated that frontoparietal feedback is reduced using two measures of directed connectivity, including the evolutional map approach (EMP) and symbolic transfer entropy (STE). It has been further demonstrated that frontoparietal feedback connectivity correlates with consciousness in humans, and such connectivity is preferentially inhibited when consciousness vanishes. Likewise, by analyzing a multichannel EEG dataset from 10 healthy volunteers, Lee et al. [65] demonstrated that the diversity of functional connectivity patterns, quantified by phase lag entropy (PLE), is decreased in propofol-induced unconsciousness. Interactions or functional connectivity between brain regions are required for proper cognitive function [66]. Alterations in functional connectivity have been proposed to underly the cognitive deficits that characterize PND. However, few studies have investigated the alterations of connectivity in PND.

### 2.5. EEG-Based Complexity

EEG-based complexity analysis is an important method to extract EEG features. High physiologic complexity indicates health, and decreased complexity is associated with poor outcomes [67,68]. Recent studies have suggested that the preoperative and intraoperative EEG-based complexity may be predictive for POD diagnosis. In a pilot study of 50 patients aged 60 years and older that underwent non-cardiac, non-neurological surgery, for whom EEG was obtained before and during anesthesia using a 32-channel EEG system, Acker et al. [28] reported that EEG-based complexity, as quantified using multi-scale entropy (MSE), was not linked to POD or attention. However, amongst the 50 patients, the average frontal EEG complexity was higher during the maintenance of anesthesia than in the preoperative wakeful eyelid closure (*p* = 0.0003). From the entropy–by-scale curve, they found that preoperative EEG MSE was higher than intraoperative EEG MSE at smaller scales, but lower at larger scales, which created an intersection point where the preoperative and intraoperative MSE curves met. In addition, in 42 patients with a single intersection point, the scale where the preoperative and intraoperative entropy curves met exhibited an inverse relationship with the change in the delirium-severity score (Spearman *ρ* = −0.31, *p* = 0.054). Thus, the average EEG complexity increased intraoperatively in the elderly but was scale-dependent. The scale where the intraoperative and preoperative entropy curves crossed could be used to predict POD. If the crossover point is established as an electrophysiological biomarker in the delirious patients, then it could have the potential to be applied in cerebral monitoring to identify patients at risk of significant neurologic complications postoperatively.

## 3. Postoperative EEG Features and PND

### 3.1. EEG Spectral Analysis

In elderly patients with POD, spectral analysis showed a decline in relative alpha and beta band power, and an increase in relative delta and theta power compared with non-delirium subjects. In a prospective clinical trial of 12 subjects after orthopedic surgery, Evans et al. [29] found greater waking EEG delta power on postoperative day 1 and less delta power during non-REM sleep as predictors of POD. However, the results were limited by the small sample size. An observational, multicenter clinical trial [30] that enrolled 159 subjects received major non-neurosurgery similarly demonstrated an association between relative EEG delta power or relative power from 1 to 6 Hz and POD, with an area under the curve of 0.75 and 0.78, respectively. As a result, relative delta power might offer an objective measurement to complement the clinical assessment of this complex, fluctuating disorder in the postoperative period [36,69]. In two different studies by Plaschke et al. [31,32], lower relative alpha and beta power and greater theta power were found in clinically delirious patients after major surgery. EEG spectral analysis appears to be a promising method of detecting POD. A shift in EEG power toward low frequencies occurs prior to obvious delirium symptoms, suggesting that it could be used as a biomarker of impending delirium.

Posterior dominant rhythm (PDR) was the first oscillatory pattern recorded in the EEG [70]. When the patient was relaxed in the eyes-closed awake state, the background of EEG was usually characterized by PDR. Adults have a stable peak frequency of PDR over the years, which can be used as a neurophysiological marker in the monitoring of cognitive impairment during acute and critical illness. More recently, Alyssa et al. [71] enrolled 60 healthy adults, randomized to either 3 h of isoflurane general anesthesia or resting wakefulness (control group), to determine whether the deviations from the baseline peak frequency of PDR were linked to cognitive function. They compared an EEG dataset recorded before the anesthetic/control period and after return of responsiveness (ROR). They found the PDR peak frequency did not change significantly over several hours in the resting wakefulness group, while the PDR peak frequency in subjects under 3 h of isoflurane general anesthesia initially declined at ROR and gradually returned toward the baseline within several hours. This research suggested that the temporal trajectory of the PDR peak frequency may be a potential biomarker for the prediction of acute cognitive dysfunction after general anesthesia. Overall, these findings support the utility of PDR peak frequency as an electrophysiological marker for serial cognitive functions.

### 3.2. EEG Emergence Trajectory

The emergence from anesthesia refers to a stage of general anesthesia characterizing the patient’s progression from the unconsciousness state to ROR [72]. It is suggested that the EEG emergence trajectory lacking alpha spindle oscillations from the end of surgery to the emergence from anesthesia may be related to POD in elderly patients. In a prospective multi-institutional observational trial, Hesse et al. [33] included 626 patients who underwent general anesthesia with planned PACU admission after elective non-cardiac surgery, monitored EEG features during the immediate post-surgical period, and statistically analyzed them for an association with delirium in the PACU. Emergence from anesthesia can take different trajectories [73]. During the immediate post-surgical period, patients who started with a delta-dominant slow-wave anesthesia (ddSWA) EEG pattern, followed by an episode of spindle-dominant slow-wave anesthesia (sdSWA) EEG and non-slow-wave anesthesia (nSWA) EEG prior to ROR, had the lowest incidence of PACU delirium. Further investigation of EEG emergence trajectories revealed that subjects whose emergence trajectories lacked spindle activity were more likely to have PACU delirium. Likewise, a published case report found that patients who lacked alpha spindle oscillations within their trajectory during emergence from anesthesia repeatedly developed PACU delirium [34]. Additionally, patients who underwent a vascular bypass procedure with prolonged POD also lacked alpha-spindle activity [35]. Hence, an emergence trajectory without alpha spindle oscillations could be identified as a predictor of PACU delirium. We could consider the emergence trajectories in EEG as a novel biomarker for predicting POD in the future.

### 3.3. EEG Connectivity

Reduced functional connectivity in the multi-channel EEG was seen in POD in the elderly. Numan et al. [36] studied functional connectivity using a 21-channel EEG in a group of patients after cardiac surgery. Functional and directed connectivity were quantified with the phase lag index (PIL) and directed phase transfer entropy (dPLE). They found that delirious patients had a significantly lower average value of PLI and a loss of posterior–anterior directionality in the alpha frequency band. The association between the reduction in functional connectivity and POD was also reported in a cross-sectional, observational, single-center clinical trial of 49 cardiac surgical patients aged 50 years or older [37]. van Dellen et al. analyzed the unbiased functional connectivity of a 21-channel EEG time series using the PLI and dPLI. The mean PLI was lower in the alpha band in delirious patients (median, 0.120; interquartile range, 0.113 to 0.138) than in cognitively normal patients (median, 0.140; interquartile range, 0.129 to 0.168; *p* < 0.01). The delta band dPLI increased towards the frontal regions more strongly in delirious patients than in non-delirious patients (F = 4.53; *p* = 0.04, and F = 7.65; *p* < 0.01, respectively). Other frequency bands were unaffected. A loss of alpha band functional connectivity and increased delta band connectivity directed toward the frontal regions characterized the EEG during delirium after cardiac surgery. The global decrease in alpha band functional connectivity during delirium was associated with the cognitive deficits, particularly attention deficits [74]. These findings are in line with the decreased functional connectivity in other disorders that impair cognitive function, such as schizophrenia and Alzheimer. Delirium therefore might be mechanistically categorized as a disconnection syndrome [37]. This functional “disconnection” prevents cortical regions of the brain from appropriately communicating and integrating information, resulting in the cognitive deficits that characterize delirium. In contrast, Tanabe et al. [38] recruited 70 surgical patients, assessed POD by CAM or CAM-ICU and measured functional connectivity by a 256-channel EEG with a debiased weighted phase lag index (wPLI) and diffusion tensor imaging (DTI). Subjects who experienced POD had increased alpha power and alpha band connectivity on the preoperative EEG, but impaired structural connectivity on preoperative DTI. These changes were compensatory responses to underlying neurodegeneration, which could be inferred from structural imaging data. Of note, the development of POD was correlated with decreased functional connectivity, which is consistent with previous studies.

In summary, POD is linked to abnormal patterns of functional and directional connectivity in the brain. EEG-based connectivity analyses can provide more insights into the interactions or functional connectivity between brain regions, which are viable methods to elucidate the neurophysiological basis of delirium. However, at present, only a few studies have used EEG to investigate functional and directional connectivity in delirium. The above-mentioned studies encourage further exploration of the utility of advanced EEG measurements as biomarkers of delirium.

### 3.4. EEG-Based Complexity

Nowadays, we have more knowledge about the association between EEG-based complexity and POD according to a new study. Here, usable EEG data recorded with a 256-channel EEG was collected from 89 participants before and after major surgery. The EEG signal complexity was estimated by calculating Lempil-Ziv Complexity (LZC). POD was diagnosed using either CAM or CAM-ICU and the severity of delirium was evaluated using the Delirium Rating Scale-98 (DRS-98). A preoperative to postoperative alteration in LZC was negatively correlated with a preoperative to postoperative alteration in DRS-98 (peak channel r^2^ = 0.199, *p* < 0.001). Likewise, in the whole brain there were statistically significant decreases in LZC that correlated with a diagnosis of delirium (peak channel 136, h*p* < 0.001) [39]. This work validated that delirium severity and delirium diagnosis are linked to reductions in LZC. In conclusion, decreased EEG-based complexity was a significant feature of POD in the elderly. Similar to other advanced EEG measures, such as functional connectivity, EEG-based complexity analysis is rarely used and investigated throughout the perioperative period.

## 4. Discussion

Anesthesiologists have made great efforts to optimize the brain health of elderly patients throughout the perioperative period [75]. EEG monitoring is now technically available in daily clinical anesthesia practice, which is typically viewed in a highly reductive manner. However, EEG could contain pivotal information about brain function and health that goes far beyond a single processed number or visually scored waveforms. Compared with the mature commercial devices used for monitoring the depth of anesthesia, multi-channel EEG processing provides richer information reflecting cognitive dysfunction in elderly patients. In this review, we summarize known intraoperative and postoperative EEG features related to PND. Clinical studies revealed that the density of the power spectrum increased in the delta band and decreased in the alpha band among elderly patients suffering from PND. Furthermore, functional connectivity and complexity between the frontal and parietal lobes appeared disturbed in PND patients. The EEG emergence trajectory of PND patients also differed from those with normal cognitive function. These EEG signatures reflect changes in brain state, as well as the integrity of certain cortical and subcortical neural circuits, during the perioperative period. As a result of the inconsistent results from trials, the issue regarding the effect of burst suppression and processed electroencephalography indices on PND remains hotly debated. Future studies may evaluate whether general anesthesia guided by these parameters derived from EEG could reduce the occurrence of this neurological complication.

Understanding the neurophysiological mechanisms is crucial to the development of novel preventive, diagnostic, therapeutic, and prognostic biomarkers of cognitive dysfunction. Some advanced EEG analyses offer insights into neurophysiology and hence can be harnessed to elucidate the pathophysiology of delirium, which will allow further investigations of EEG measures as biomarkers of cognitive dysfunction. Establishing real-time and relatively affordable perioperative electrophysiological biomarkers can facilitate the early identification of patients that are potentially at an increased risk of significant postoperative neurologic complications. Unfortunately, the existing observations do not support a causal relationship between the above-mentioned EEG features and postoperative cognitive dysfunction. One limitation is that we failed to accurately interlock the EEG data with cognitive performance, such as the usage of the alpha/beta ratio in attention assessment. We therefore hope to establish an objective multi-channel EEG-based strategy relying on the specific and event-related EEG activity in the elderly, such as the weakening of the alpha oscillation or the disruption of phase amplitude coupling during general anesthesia, which would provide a quantitative index as a predictor of PND in the future. Targeted treatment for dysfunction in certain cognitive domains will facilitate the early recovery of cognitive function, reduce the incidence of short-term and long-term complications, and improve the overall quality of life in elderly patients.

## Figures and Tables

**Table 1 brainsci-12-01073-t001:** Studies about intraoperative EEG features related to PND.

Study	Design	Sample Characteristics	Surgery	Primary Anesthetic Maintenance Drug	Cognitive Function Tool	EEG Set-Up	Summary Finding
Chan et al. [9]	RCT	*n* = 921 BIS guided anesthesia *n* = 462 Routine care anesthesia *n* = 459 Age ≥60	Elective major surgery	Propofol	At 3 months after surgery A neuropsychology battery of tests	Intra. BIS	Without BIS-guided anesthesia
Evered et al. [10]	RCT	*n* = 515 BIS 50 *n* = 253 BIS 35 *n* = 262 Age ≥60	Major surgery	Volatile anesthetic	Post. 3D-CAM/CAM-ICU	Intra. BIS	Deep anesthesia
Wildes et al. [11]	RCT	*n* = 1213 Delirium *n* = 297 Non-delirium *n* = 916 Age ≥60	Major surgery	Volatile anesthetics	Post. CAM/CAM-ICU/chart review	Intra. BIS	n.s. processed EEG indices
Cartailler et al. [12]	POS	*n* = 38 Cognitive decline *n* = 18 No cognitive decline *n* = 20 Age = 69 (10.6)	Orthopedic surgery/neuroradiology intervention	Propofol	Pre. MoCA	During induction of general anesthesia, SedLine brain function monitor	↑ TAD ↓ Alpha power
Giattino et al. [13]	POS	*n* = 50 Age = 69 (5.5)	Non-cardiac, non-neurologic surgery	Propofol/isoflurane	Pre. neurocognitive test battery	Intra. 32-channel EEG/ BIS	↓ Alpha power
Gutiérrez et al. [14]	RS	*n*= 35 Low MoCA group *n* = 12 High MoCA group *n* = 23 Age ≥60	Elective major abdominal surgery	Sevoflurane or desflurane	Pre. MoCA	Intra. 16-channels EEG	↓ Alpha-beta power ↓ Alpha peak frequency ↓ Alpha band coherence ↑ PAC
Touchard et al. [15]	POS	*n* = 42 CD *n* = 17 Non-CD *n* = 25 Age = 59.4 (18.8)	Interventional neuroradiology or orthopedic surgery	Propofol	Pre. MoCA	Intra. Sedline brain function monitor	↓ Alpha power
Koch et al. [16]	POS	*n* = 38 Lower cognitive *n* = 14 Normal cognitive *n* = 24 Age = 71.8 (4.6)	Elective surgery	Propofol/sevoflurane/desflurane	Pre. 6 neuropsychological tests	Intra. SedLine Root Monitor	↓ Alpha power ↓ Alpha -peak power
Gutierrez et al. [17]	POS	*n* = 30 PD/PSSD *n* = 13 Normal CAM *n* = 17 Age = 72.1 (7.0)	Elective major abdominal surgery	Sevoflurane/desflurane	Post. CAM	Intra. 16-channel EEG	↓ Alpha power
Fritz et al. [18]	POS	*n* = 619 Delirium *n* = 162 Non-delirium *n* = 457 Age = 62 (14)	Non-neurologic surgery	Propofol, sevoflurane, desflurane, or a combination of these agents, with or without nitrous oxide	Post. CAM-ICU	Intra. 1-channel BIS	↑ Time in burst suppression
Momeni et al. [19]	POS	*n* = 1515 POD *n* = 303 Non-POD *n* = 1201 POCD *n* = 270 Non-POCD *n* = 1080 Age = 68 (range 58–77)	First or redo cardiac surgery/TAVI	Sevoflurane	Post. chart review method and MMSE	Intra. 2 bilateral channels EEG	↑ Time in burst suppression
Soehle et al. [20]	POS	*n* = 81 Delirium *n* = 26 Non-delirium *n* = 55 Age = 72.9 (6.2)	Cardiac surgery	Isoflurane	Post. CAM-ICU flowchart	Intra. 2-channel BIS	↑ Time in burst suppression
Pedemonte et al. [21]	RS	*n* = 141 Burst suppression *n* = 60 No burst suppression *n* = 81 Age ≥60	Cardiac surgery	Isoflurane	Post. long version of the CAM	Intra. SedLine monitor	Experience of burst suppression
Lele et al. [22]	RS	*n* = 112 Delirium *n* = 10 Non-delirium *n* = 102 Age = 59.8 (18.8)	Spine instrumentation surgery	Propofol	Post. CAM	Intra. 4-channel EEG	↑ Time in burst suppression
Fritz et al. [23]	RS	*n* = 618 Delirium *n* = 162 Non-delirium *n* = 456 Age = 62 (range 18–92)	Elective surgery	Volatile anesthetic	Post. CAM-ICU	Intra. 1-channel BIS	Experience of burst suppression at relatively lower concentrations of volatile anesthetic
Wildes et al. [11]	RCT	*n* = 1213 Delirium *n* = 297 Non-delirium *n* = 916 Age ≥60	Major surgery	Volatile anesthetic	Post. CAM/CAM-ICU/chart review	Intra. BIS	n.s. time in burst suppression
Tang et al. [24]	RCT	*n* = 201 Delirium *n* = 37 Non-delirium *n* = 164 Age = 72 (5)	Major elective, non-cardiac surgery	Inhaled and intravenous agents	Post. CAM	Intra. SedLine Brain Function Monitor	n.s. time in burst suppression
Deiner et al. [25]	RS	*n* = 77 POCD *n* = 21 Non-POCD *n* = 56 Age>68	Major non-cardiac surgery	Propofol, sevoflurane	3 months after surgery, a neuropsychological battery plus MMSE and CAM	Intra. BIS	↓ Time in burst suppression
Shao et al. [26]	RS	*n* = 155 Age = 48.69 (18.57)	Elective surgery	Propofol/sevoflurane		Intra. SedLine Brain Function Monitor	Lower frontal alpha power is strongly associated with a higher propensity for burst suppression
Plummer et al. [27]	RS	*n* = 138 Age = 69.1 (9.1)	Cardiac surgery with CPB	Isoflurane		Intra. 4-channel EEG	↓ Intra-operative power within the alpha and beta range was linked to susceptibility to burst suppression
Acker et al. [28]	POS	*n* = 50 Decrease in attention score *n* = 18 No decrease in attention score *n* = 32 Age = 68.8 (5.4)	Non-cardiac, non-neurological surgery		Post. 3D-CAM	Intra. 32-channel EEG	↓ Crossover point

Note: ↓ denotes decrease; ↑ denotes; n.s. denotes non-significant association between PND and EEG parameters. RS—retrospective study; POS—prospective observational study; RCT—randomized controlled trial; Pre.—preoperative; Intra.—intraoperative; Post.—postoperative; TAD—transient amplitude decreased; CAM—confusion assessment method; CAM-ICU—confusion assessment method for intensive care unit; MoCA—Montreal Cognitive Assessment; MMSE—Mini-mental State Examination; BIS—bispectral index monitoring; EEG—electroencephalogram; POCD—postoperative neurocognitive disorder; TAVI—Transcatheter Aortic Valve Implantation; CPB—cardiopulmonary bypass; PAC—peak-max phase–amplitude coupling; CD—cognitive decline.

**Table 2 brainsci-12-01073-t002:** Studies about postoperative EEG features related to PND.

Study	Design	Sample Characteristics	Surgery	Primary Anesthetic Maintenance Drug	Cognitive Function Tool	EEG Set-Up	Summary Finding
Evans et al. [29]	POS	*n* = 12 Delirium *n* = 3 Non-delirium *n* = 9 Age = 66.8 (8.2)	Orthopedic surgery		Post. CAM-ICU	Post. PSG	↑ Waking delta power ↓ Delta during non-REM sleep
Numan et al. [30]	POS	*n* = 159 Delirium *n* = 26 Non-delirium *n* = 104 Age = 76.9 (6.2)	Non-neurological, major surgery		Post. CAM-ICU	Post. 3-channel EEG	↑ Delta power
Plaschke et al. [31]	POS	*n* = 37 Delirium *n* = 17 Non-delirium *n* = 20 Age = 63.6 (11.6)	Elective surgery		Post. CAM-ICU	Post. 16-channel EEG	↑ Theta power ↓ Alpha power ↓ Beta power
Plaschke et al. [32]	POS	*n* = 114 Delirium *n* = 32 Non-delirium *n* = 82 Age = 69 (8.9)	Open-heart cardiac surgery		Post. CAM-ICU	Bilateral 4-channel EEG	↑ Theta power ↓ Alpha power
Hesse et al. [33]	POS	*n* = 626 PACU delirium *n* = 125 Non-PACU delirium *n* = 501 Age = 57 (range 44–68)	Non-emergency non-cardiac surgery	Sevoflurane/isoflurane/propofol/desflurane	Post. CAM-ICU	Emergence period SedLine Brain Function Monitor	Lacking significant spindle power
Kreuzer et al. [34]	Case report	Age = 37 Sex = female Repeatedly developed PACU delirium Pedestrian versus car accident	Required 22 surgeries	Sevoflurane/propofol	After 10 of her 22 surgeries CAM	Emergence period SedLine	Lacking alpha spindle oscillations containing trajectory
Whalin et al. [35]	Case report	Age = 56 Sex = female With prolonged POD	A vascular bypass procedure	Sevoflurane	Post. CAM-ICU	Emergence Frontal EEG	Lacking alpha spindle activity
Numan et al. [36]	POS	*n* = 58 Delirium *n* = 18 No delirium *n* = 20 recovering from anesthesia *n* = 20 Age = 75.3 (6.4)	Cardiac surgery	Sevoflurane, isoflurane	Post. DSM-Ⅳ-R, CAM-ICU	Post. 21-channel EEG	↑ Delta power ↓ Alpha power ↓ Functional connectivity
Van Dellen al. [37]	Cross-sectional	*n* = 49 Delirium *n* = 25 no Delirium *n* = 24 Age = 75.1 (7.6)	Cardiac surgery		Post. DSM-Ⅳ and CAM-ICU	Post. 21-channel EEG	Loss of alpha band functional connectivity ↑Delta band connectivity
Tanabe et al. [38]	POS	*n* = 70 Delirium *n* = 22 Non-delirium *n* = 48 Age > 65	Major surgery		Post. CAM/CAM-ICU	Pre. 256-channel EEG	↑ Alpha power ↑ Alpha band connectivity
Tanabe et al. [39]	POS	*n* = 89 Delirium *n* = 30 Non-delirium *n* = 59 Age > 65	Major surgery		Post. CAM or CAM-ICU	Pre. and post. 256-channel EEG	↓ Complexity

Note: ↓ denotes decrease; ↑ denotes increase n.s. denotes non-significant association between PND and EEG parameter. RS—retrospective study; POS—prospective observational study; Pre.—preoperative; Intra.—intraoperative; Post.—postoperative; CAM—confusion assessment method; CAM-ICU—confusion assessment method for intensive care unit; DSM-Ⅳ—diagnostic and statistical manual of mental disorders fourth edition; DSM-Ⅳ-R—diagnostic and statistical manual of mental disorders fourth edition text revision; BIS—bispectral index monitoring; EEG—electroencephalogram; POD—postoperative delirium; PACU—post anesthesia care unit; REM—rapid eye movement.

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
