# Peer review of "Electroencephalogram Features of Perioperative Neurocognitive Disorders in Elderly Patients: A Narrative Review of the Clinical Literature"

_brainsci, 2022, doi:10.3390/brainsci12081073_

Round 1

Reviewer 1 Report

General. 

 This review article has appropriate content although a couple of large relevant studies are not mentioned (see below).

I struggled to read the article because the quality of English was not very good throughout.  At times this impacted on interpretation/meaning as well as readabilty. I feel this needs to be addressed dramatically to bring the review up to a publishable standard.

Some specific comments:

Abstract

Unclear terms such as  ‘mature’ indicators of depth of anaesthesia – ‘processed EEG  indices’ or ‘proprietary’ indices – is more common terminology. Alternatively ‘correlates’ or ‘markers’ might be better terms

Introduction

Again the English is not always clear – sometimes is too colloquial otherwise not terribly fluent / incorrect plurals etc

Section 2.1.

The ENGAGES trial  primarily examined avoidance of burst suppression rather than pEEG indices.  Some important large studies such as CODA and BALANCED are omitted and should be mentioned

Section 2.2 lm 144 – unclear what the two correlation coefficients relate to

Author Response

Dear Editors and Reviewers:
Thank you for your letter and for the reviewers’ comments concerning our manuscript entitled “Electroencephalogram features of perioperative neurocognitive disorders in elderly patients: a narrative review of clinical literature”. (ID: brainsci-1815846). Those comments are all valuable and very helpful for revising and improving our paper, as well as the important guiding significance to our research. We have studied comments carefully and have made the correction which we hope to meet with approval. The revised portion is marked in red in the paper. The main corrections in the paper and the responses to the reviewer’s comments are as flowing:

Response to Reviewer1 Comments

Point 1:

General. 

This review article has appropriate content although a couple of large relevant studies are not mentioned (see below).

I struggled to read the article because the quality of English was not very good throughout.  At times this impacted on interpretation/meaning as well as readabilty. I feel this needs to be addressed dramatically to bring the review up to a publishable standard.

Response 1:

We are very sorry for our negligence of a couple of large relevant studies. Considering the Reviewer’s suggestion, we have again reviewed recent studies about intraoperative and postoperative raw features of EEG data related to PND in elderly patients. Some important large studies such as CODA and BALANCED are added.

Our manuscript has undergone extensive English revisions during revision. The revised portion is marked in red in the paper.

Point 2:

Abstract

Unclear terms such as ‘mature’ indicators of depth of anaesthesia – ‘processed EEG indices’ or ‘proprietary’ indices – is more common terminology. Alternatively, ‘correlates’ or ‘markers’ might be better terms

Response 2:

Line 16, the statements of ‘mature’ were corrected as ‘common’. Nowadays, common indicators of depth of anaesthesia refers to ‘processed EEG indices’ or ‘proprietary’ indices, such as BIS, Narcotrend and etc.

Point 3:

Introduction

Again the English is not always clear – sometimes is too colloquial otherwise not terribly fluent / incorrect plurals etc

Response 3:

Our manuscript has undergone extensive English revisions during revision. The revised portion is marked in red in the paper.

Point 4:

Section 2.1

The ENGAGES trial primarily examined avoidance of burst suppression rather than pEEG indices.  Some important large studies such as CODA and BALANCED are omitted and should be mentioned

Response 4:

We carefully read the article, found that the ENGAGES trial was designed to investigate whether reducing anesthetic administration and minimizing EEG suppression during surgical anesthesia decreased the incidence of POD. As Reviewer said that the ENGAGES trial primarily examined avoidance of burst suppression rather than pEEG indices. Despite ENGAGES has been criticized on several levels, the ENGAGES trial indeed demonstrated that pEEG monitors did not decrease the incidence of POD in elderly patients. We have re-written this part according to the Reviewer’s suggestion.

Some important large studies such as CODA and BALANCED was added.

Point 5

Section 2.2 lm 144 – unclear what the two correlation coefficients relate to

Response 5:

Im 174 (revised version), the former correlation coefficient (rs = 0.593, p = 0.022) refers correlation between intraoperative frontal alpha power and preoperative cognitive index score in patients whose EEG signals from 32-channel EEG monitoring. The latter correlation coefficient (rs= 0.338, p = 0.047) refers correlation between intraoperative frontal alpha power and preoperative cognitive index score in patients whose EEG signals from BIS monitoring.

Special thanks to you for your good comments.

We tried our best to improve the manuscript and made some changes in the manuscript.  These changes will not influence the content and framework of the paper. And here we did not list the changes but marked in red in revised paper.
We appreciate for Editors and Reviewers’ warm work earnestly and hope that the correction will meet with approval.
Once again, thank you very much for your comments and suggestions.

Reviewer 2 Report

Dear authors,

I read your review with interest. It is an important topic in the matter. During the last decades EEG monitors have been introduced in the operating room. These monitors allow us to know the effect of general anesthetic drugs on the brain. As the authors mentioned in the review, EEG-guided dosing of general anesthetics could prevent postoperative cognitive complications.

In this review, the authors summarize the evidence on whether some EEG features in the intraoperative and postoperative period indicate or predict postoperative neurocognitive disorders. The review is well written and covers the main ideas published to date. However, the authors must include some articles and discuss some relevant topics in this review.

- In section 2.1. Processed EEG indices: I suggest including and discussing the results of Evered et al. British J Anaesth. 2021 November; 127 (5): 704-712. Also, there are more meta-analyses showing that pEEG-guided anesthesia prevents POD, I suggest citing these meta-analyses. In fact, the only trial that failed to replicate the results was ENGAGE, which is admitted to have several problems, including that there was no clinically relevant difference in general anesthetic concentration between groups. In addition, the prevalence of POD was higher than in other trials because the patients were highly susceptible to developing POD (prior POD, benzodiazepines, comorbidity, among others). Therefore, pEEG monitors probably do not prevent POD in highly susceptible patients, but this does not mean that EEG-based anesthesia guidance does not prevent POD.

- In section 2.2.2. Alpha power and preoperative cognitive decline. TAD is Transient amplitude decreased.

- In the same section, authors should include citations from Touchard C, et al. Front Aging Neurosci 2020; 12:593320; and Gutierrez et al. Analgesic Anesth. July 1, 2021; 133 (1): 205-214. Both articles demonstrate that preoperative cognitive status is associated with the sensitivity of the effect of general anesthetics. Thus, patients with some degree of cognitive impairment are more sensitive to the effect of general anesthetics, expressed in a greater slowing of EEG activity at the same anesthetic dose than a cognitively more robust patient. This raises the question of whether the intraoperative EEG really allows the detection of the most susceptible patients, rather than allowing the dosage to be changed and reducing the possibility of developing a cognitive alteration in the postoperative period.

- In section 3.4. Complexity based on EEG, the reference is cited far below.

- It could be useful to include a summary outline. For example, three stages: pre, intra and postoperative. And down tick the different EEG features associated with delirium or other NCP.

Author Response

Point 1

In section 2.1. Processed EEG indices: I suggest including and discussing the results of Evered et al. British J Anaesth. 2021 November; 127 (5): 704-712. Also, there are more meta-analyses showing that pEEG-guided anesthesia prevents POD, I suggest citing these meta-analyses. In fact, the only trial that failed to replicate the results was ENGAGE, which is admitted to have several problems, including that there was no clinically relevant difference in general anesthetic concentration between groups. In addition, the prevalence of POD was higher than in other trials because the patients were highly susceptible to developing POD (prior POD, benzodiazepines, comorbidity, among others). Therefore, pEEG monitors probably do not prevent POD in highly susceptible patients, but this does not mean that EEG-based anesthesia guidance does not prevent POD.

Response 1:

We are very sorry for our negligence of a couple of large relevant studies and meta-analyses showing that pEEG-guided anaesthesia prevents POD, we have supplemented these relevant studies and meta-analyses according to the Reviewer’s suggestion. We include and discuss the results of Evered et al. British J Anaesth. 2021 November; 127 (5): 704-712.Punjasawadwong et al. Cochrane Database Syst Rev. 2018 May 15;5(5):CD011283. MacKenzie et al. Anesthesiology. 2018 Sep;129(3):417-427. Sun et al. Anesth Analg. 2020 Sep;131(3):712-719.The revised portion is marked in red in the paper.

Point 2:

In section 2.2.2. Alpha power and preoperative cognitive decline. TAD is Transient amplitude decreased.

In the same section, authors should include citations from Touchard C, et al. Front Aging Neurosci 2020; 12:593320; and Gutierrez et al. Analgesic Anesth. July 1, 2021; 133 (1): 205-214. Both articles demonstrate that preoperative cognitive status is associated with the sensitivity of the effect of general anesthetics. Thus, patients with some degree of cognitive impairment are more sensitive to the effect of general anesthetics, expressed in a greater slowing of EEG activity at the same anesthetic dose than a cognitively more robust patient. This raises the question of whether the intraoperative EEG really allows the detection of the most susceptible patients, rather than allowing the dosage to be changed and reducing the possibility of developing a cognitive alteration in the postoperative period.

Response 2:

In section 2.2.2. the statements of ‘transient alpha power decrease (TAD)’ were corrected as ‘transient amplitude decreased (TAD)’.

As reviewer suggested that we included citations from Touchard C, et al. Front Aging Neurosci 2020; 12:593320; and Gutierrez et al. Analgesic Anesth. July 1, 2021; 133 (1): 205-214. in the review.

We read the two articles carefully according to the Reviewer’s comments. Patients who have low cognitive performance are more sensitive to the effect of general anesthetics and have greater slowing of EEG activity intraoperatively. Thus, for patients with poorer preoperative cognitive function, EEG-guided dosing of general anaesthetics could probably decrease the incidence of postoperative cognitive complication.

We think slowing of EEG activity such as decreased alpha power may be an electrophysiological biomarker of cognitive impairment. Above-mentioned studies render further investigations of EEG measures as biomarkers of cognitive dysfunction.

Point 3:

In section 3.4. Complexity based on EEG, the reference is cited far below.

Response 3:

We are very sorry we do not find other relevant studies about the association between EEG-based complexity and PND. Nowadays, there are few studies on the relationship between the EEG-based complexity and PND.

In this article, EEG data were collected using saline-cap–based high-density-EEG with 256 channels. Complexity of the EEG signal fades proportionately to delirium severity implying reduced cortical information. Despite there are some limitations, such as modest size (n=89) and observational study, the study suggests EEG-based complexity indices as a promising novel pathological biomarker

of the delirium state. The above-mentioned study also encourages further investigation of the utility of advanced EEG measurements as biomarkers of delirium for diagnosis and prognosis.

Point 4:

It could be useful to include a summary outline. For example, three stages: pre, intra and postoperative. And down tick the different EEG features associated with delirium or other NCP.

Response 4:

Including a summary outline is useful according to the Reviewer’s suggestion. In this review, we summarize the evidence on whether some EEG features in the intraoperative and postoperative period predict PND. Our summary outline is from intraoperative to postoperative period.

In the intraoperative period, lower BIS value, increased TAD, decreased alpha power, presence of burst suppression, longer time in burst suppression and the scale at which preoperative

and intraoperative complexity is equal are all associated with PND.

In the postoperative period, increased delta and theta power, decreased alpha and beta power, the EEG emergence trajectory lacking significant spindle activity, decreased alpha band functional connectivity, increased delta band functional connectivity and decreased complexity are all associated with PND.

Above-mentioned EEG features are presented in table1 (in the intraoperative period) and table2 (in the postoperative period). These EEG signatures reflect changes in brain state, as well as the integrity of certain cortical and subcortical neural circuits, during the perioperative period.

We mainly focus on the EEG features intraoperatively and postoperatively, which is affected by some anesthetics and other drugs. As Koch et al. (Dement Geriatr Cogn Disord. 2019;48(1-2):83-92.) mentioned, their studies indicate a higher sensitivity of physiological age-related changes in cerebral activity seen after GABA-A activation induced by general anesthesia compared to baseline

cerebral activity. Therefore, EEG features in the intraoperative period may indicate or predict postoperative neurocognitive disorders better than baseline. We will discuss the preoperative EEG features related PND in another article.

Special thanks to you for your good comments.

We tried our best to improve the manuscript and made some changes in the manuscript.  These changes will not influence the content and framework of the paper. And here we did not list the changes but marked in red in revised paper.
We appreciate for Editors and Reviewers’ warm work earnestly and hope that the correction will meet with approval.
Once again, thank you very much for your comments and suggestions.

Reviewer 3 Report

Beautiful narrative review

The topic is very important: EEG  intraoperative monitoring is the future in Anaesthesia and Intensive Care, like a normal ECG

I think the authors have discussed the topic well, and I have no concerns or suggestions.

Is very important the digression about Burst Suppression

Well done

Author Response

Point 1:

Is very important the digression about Burst Suppression

Response 1:

There has since been considerable interest in examining both the physiological underpinnings and clinical implications of EEG burst suppression. Nowadays, there are large number of studies on the association between the EEG burst suppression and PND. By quantifying real-time EEG burst suppression, it was shown that burst suppression is an independent risk factor for the development of PND (Fritz et al. Anesth Analg. 2016 Jan;122(1):234-42.). Further, the longer the patients remain in intraoperative EEG suppression, the higher the likelihood of developing POD and delayed cognitive recovery. Therefore, EEG suppression is an important biomarker of PND, it is an important part in the review.

Special thanks to you for your good comments.

We tried our best to improve the manuscript and made some changes in the manuscript.  These changes will not influence the content and framework of the paper. And here we did not list the changes but marked in red in revised paper.
We appreciate for Editors and Reviewers’ warm work earnestly and hope that the correction will meet with approval.
Once again, thank you very much for your comments and suggestions.